# A Configurable IP Core for Calculating the Integer Square Root for Serial and Parallel Implementations in FPGA

Vladimir Matyukha [1], Sergey Voloshchuk [1] and Sergey Mosin [1,2,*]

1   LabSystems LLC, Fedoseeva Str. 5, 600000 Vladimir, Russia; vladimir.matyuha@lab-systems.ru (V.M.); sergey.voloshchuk@lab-systems.ru (S.V.)
2   Institute of Computational Mathematics and Information Technologies, Kazan Federal University, Kremlyovskaya Str. 18, 420008 Kazan, Russia
*   Correspondence: smosin@ieee.org; Tel.: +7-492-260-47-08

**Abstract:** The development of digital technologies is in many ways associated with an improvement of integrated technologies, microelectronic components, and the capabilities of hardware acceleration of the most computationally complex operations. Field-programmable gate arrays (FPGAs) are actively used for prototyping or the small-scale production of special purpose digital signal processing (DSP) devices. The implementation of DSP algorithms is variative in nature and affects important indicators of a produced device, such as the accuracy of the numerical solution, performance, structural/functional complexity, etc. The architectural features of the FPGA can be used for choosing an effective DSP algorithm in the form of solving the multicriteria discrete optimization problem. This paper analyzes and selects an effective algorithm for calculating the integer square root, which is one of the most frequently used digital signal processing operations. A behavioral model based on a non-restoring algorithm is presented. The SystemVerilog description of the module for calculating the square root, presented in the form of a universal configurable IP core, has been developed and synthesized. The configuration allows one to change the width of the input data bus and select the serial or parallel processing mode for scalar or vector data. The results of testing and comparison of the obtained characteristics with the corresponding Xilinx Cordic IP core are presented. The field test of the proposed IP core implemented in the Xilinx FPGA SOC xc7z045ffg900-2 has demonstrated the gain in the maximum system frequency at 174 MHz in the sequential mode with a 48-bit input bus and 169 MHz in the pipelined mode at a reduction of both the structural complexity and the number of used FPGA internal resources in comparison with the Xilinx Cordic IP core.

**Keywords:** integer square root; non-restoring algorithm; FPGA design; pipelined data processing

## 1. Introduction

The operation of square root calculation (OSRC) is often found in DSP algorithms, for example, when calculating the modulus of a complex number, root-mean square (RMS) value, and standard deviation, etc. The OSRC is widely used in telecommunication, including in 5G and beyond 5G mobile systems, for estimating the signal amplitude based on IQ constituents and bit-error ratio (BER). Hardware implementation can significantly reduce the time spent on performing this operation.

The programmable logic gate array's integrated circuits (FPGA) are widely used as a universal basis for implementation of DSP and specialized digital computers, emphasizing on performance and improvement of area overheads. For instance, a specialized DSP processor is proposed, implemented, and estimated on a Xilinx Virtex-5 XC5VSX50T FPGA chip [1]. The overall performance there was improved by an average of nine times in an FIR Filter and Matrix Multiplication benchmark. A tool for mapping graphs of add/sub/mul nodes to DSP blocks on Xilinx FPGAs, ensuring maximum throughput, is developed and presented in ref. [2]. An approach to strike a favorable balance between utilization of the

FPGA on-chip memory, logic, and DSP resources for convolution computation, which leads to transformation of the accelerator design space and relaxes the pressure on the required DSP resources, is proposed in ref. [3]. The multiplier or addition blocks are considered in ref. [4], which enable the DSP block to directly support the multi-operands addition operation with high performance and effective use of the chip area. The use of hardware description languages for the development and implementation of complex functional devices in the FPGA basis is described in refs. [5,6]. The development of an approximate square root circuit is presented in ref. [7]. The final solution provides low latency and power dissipation.

The aim of this paper deals with the development and verification of the OSRC implementation module as an IP core for subsequent reuse according to the design reuse paradigm. An important task when implementing an FPGA-based design is the optimal use of the chip's internal hardware resources. Modern FPGAs provide an opportunity to use not only standard logic for a project implementation but also more complex internal functional blocks, for example, look-up-tables (LUTs), multipliers, block memory, etc. Knowledge of the architectural features of FPGAs can be used when choosing an effective algorithm for performing the OSRC ($n$*) in the form of solving a multicriteria discrete optimization problem

$$\min_{n \in D} F(n) = F(n^*) \tag{1}$$

where

$D = \{1, \ldots, N\}$ is the admissible range of variable parameters for $N$ algorithms;

$n \in D$ is the number of the OSRC implementation algorithm from the set of considered options $A = \{a_n\}_{n=1}^{N}$;

$F(n) = (f_1(n), f_2(n), f_3(n))$ is a vector criterion of optimality including:

- An accuracy of square root calculation ($f_1(n) = -m_{1n} \to \min_n$);
- A time of square root calculation ($f_2(n) = m_{2n} \to \min_n$);
- A hardware implementation complexity of the OSRC ($f_3(n) = m_{3n} \to \min_n$);

$M = [m_{in}]$ is a $3 \times N$ matrix of initial parameters for the set of considered options $A$.

There are many algorithms for implementing the OSRC for integers and real numbers based on the use of table functions, approximations, iterative numerical methods, or polynomial approaches. However, not all algorithms are equally applicable for efficient implementation in the FPGA basis. An algorithm in ref. [8] uses simple arithmetic operations. The error of the algorithm decreases with increasing iterations. The disadvantages of the algorithm include suboptimal resource consumption with the required accuracy of calculations. A 16-bit input bus implementation requires 800 logic gates and 600 flip-flops. An algorithm in ref. [9] uses a polynomial approach to implementation using digital signal processors (DSPs) and block memory (BRAM). A 16-bit input bus implementation requires 177 logic gates, 176 flip-flops, 2 BRAMs, and 2 DSPs. An algorithm in ref. [10] uses a distributed memory in which precalculated values of the square root are stored. As the memory size decreases, the number of calculation iterations increases. For a 16-bit input bus design, 325 pairs of logic blocks are required, including a lookup table and a flip-flop (LUT-FF). An iterative algorithm for calculating the square root in the CORDIC (coordinate rotation in a digital compute) basis is presented in ref. [11] using shift and addition operations. This apparatus is often used to calculate trigonometric and special functions as well. The non-restoring square root algorithm is presented and discussed in refs. [12–16]. The algorithm is based on sequential consideration of a pair of the operand's bits, so at each iteration step one bit of the result is formed, and the number of iterations is finite and deterministically equal to half the length of the operand to calculate the integer part of the square root. An additional iteration is used to calculate the rounding bit. The algorithm uses simple arithmetic operations and does not require DSP and/or BRAM. A 16-bit input bus implementation takes about 200 gates and flip-flops.

The analysis of the existing algorithms for performing the OSRC showed that the non-restoring algorithm has the best characteristics for implementation in the FPGA basis, taking

into account criterion (1). This paper proposes a behavioral model and implementation of an IP core for integer square root calculation with customizable restrictions: input data width from 8 to 128 bits, the number of resources used for a 16-bit bus should not exceed 0.1% of the chip area (with focus on the Xilinx FPGA SOC xc7z045ffg900-2), and a calculation of the square root with integer precision in a predictable number of clock cycles proportional to the width of the input bus. Both the model and the SystemVerilog description were verified. A field full-scale testing of the synthesized IP core and implementation on the Xilinx FPGA SOC xc7z045ffg900-2 basis were conducted. A comparative analysis of the obtained characteristics for the implemented IP core with characteristics of the Xilinx IP core based on the CORDIC algorithm was performed.

## 2. Integer Square Root Models

### 2.1. CORDIC-Based Model

The CORDIC (coordinate rotation digital computer) method was originally proposed for calculating trigonometric functions and coordinate transformation operations [11] and later was extended to exponential and logarithmic functions, including square root extraction. The main attraction of the CORDIC method lies in the use of operations in the calculation of complex functions that are implemented through a combination of simple steps of addition and shift-operations that require minimal resources for hardware implementation.

The basic CORDIC algorithm for calculating the square root uses the operations of multiplication, addition/subtraction, and shift, while multiplication, and hence the entire set of operations involved in the hardware implementation, can be represented by decomposition of addition and shift only. The algorithm is iterative, allowing for a finite number of steps to obtain the result.

---

**Basic CORDIC algorithm** of square root calculation

---

function [*data_out*] = cordic_SQRT(*data_in*)
% CORDIC square-root calculation
*N* = ceil(log2(*data_in*)/2);                    % number of bits for the result value
*Base* = 2 ˆ *N*;                                  % *Base* assignment
*data_out* = 0;                                    % the result value initialisation
for *m* = 1: *N* + 1,                              % iterative calculation of the result value
*data_out* = *data_out* + *Base*;                  % add
if (*data_out* * *data_out*) > *data_in*,          % mul and comparison
*data_out* = *data_out* − *Base*;                  % correction
end
*Base* = bitshift(*Base*, −1);                     % div by 2 (shift left by 1 bit)
end

---

The Xilinx LogiCORE™ IP CORDIC library includes IP cores for executing complex functions based on the CORDIC method [17]. There are two architectural configurations for the CORDIC core: word serial, with multiple-cycle throughput and a smaller silicon area (Figure 1a), and parallel, with single-cycle data throughput and large silicon area (Figure 1b).

The CORDIC algorithm requires approximately one Shift-AddSub operation for each bit of accuracy. A CORDIC core with a parallel architectural configuration implements these Shift-AddSub operations in parallel using an array of Shift-AddSub stages. A parallel CORDIC core with $N$ bit output width has a latency of $N$ cycles and produces a new output every cycle. A CORDIC core implemented with the word serial architectural configuration implements these Shift-AddSub operations serially, using a single Shift-AddSub stage and feeding back the output. A word serial CORDIC core with $N$ bit output width has a latency of $N$ cycles and produces a new output every $N$ cycle [18].

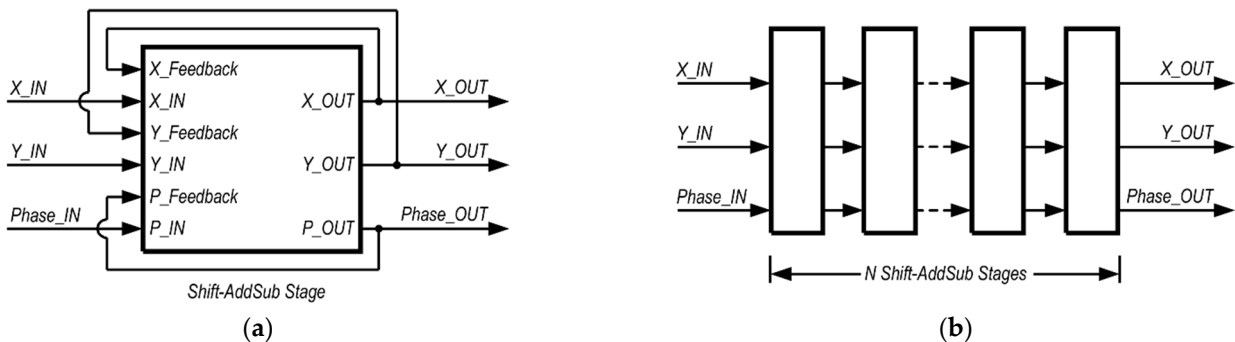

**Figure 1.** The Xilinx LogiCORE™ IP CORDIC architectural configurations: (**a**) word serial; (**b**) parallel.

It should be noted that the Xilinx IP core for the square root calculation is only synthesized for the parallel architecture.

### 2.2. Proposed Model Based on a Non-Restoring Algorithm

The construction of a behavioral model is based on a non-restoring algorithm for calculating the integer square root, which uses at each step the simplest operations for hardware implementation such as the logical shift and subtraction.

The essence of the algorithm is the following. The original number in binary representation is split into pairs of bits, and the calculation starts with the most significant pair of bits. Two numbers are obtained at each iteration: the minuend and the subtrahend. The minuend is formed by concatenating the remainder from the previous iteration and a pair of bits from the input number: the most significant pair of bits is the remainder; the lower pair of bits is a pair of bits from the input number. The subtrahend is formed by concatenation of the already-calculated bits of the result of extracting the square root and the constant $01_2$: the number calculated at the previous iteration is put in the high order bits, and $01_2$ is put in the lower bits. If the difference of these two numbers is greater than or equal to zero, the found bit of the result is one, and otherwise it is zero. If the difference is greater than or equal to zero at the current iteration, the difference from the current iteration is taken as the remainder. If the difference is less than zero, the remainder from the previous iteration should be taken as the remainder. The algorithm accurately calculates the next bit of the integer square root result and the remainder at each iteration.

Thus, the presented algorithm allows one to obtain the result of integer square root calculation and has the following computational complexity (2):

$$O(data\_out\_size + 1) \tag{2}$$

$$data\_out\_size = ceil\left(\frac{floor(log_2(data\_in)) + 1}{2}\right)$$

where *data_in* is the input number, and *data_out_size* is the length of the output number in bits.

So, the proposed algorithm provides the result of integer square root in nine steps for a 16-bit input number, in 17 steps for a 32-bit input number, in 25 steps for a 48-bit input number, and so on.

A behavioral model was developed in the MATLAB tool as a function based on the algorithm described above, where an input variable *data_in* is the input value, and *data_out* is the returned result.

| Behavioral model for operation of integer square root calculation |
|---|

function [*data_out*] = com_sqrt_calculator(*data_in*)
*data_out* = 0;
if *data_in* ~= 0
*data_out_size* = ceil((floor(log2(*data_in*)) + 1)/2);       % length of output value (in bit)
*data_bin* = de2bi(*data_in*, 2 * *data_out_size*, 'left-msb');       % decimal to binary transformation
*data_bin* = [*data_bin* 0 0];
*register* = 0;       % register initialization
for *i* = 1: *data_out_size* + 1       % iterative calculations
*register* = 4 * *register* + bi2de(*data_bin*((*i* − 1) * 2 + 1: (*i* − 1) * 2 + 2), 'left-msb');
if (*register* − 4 * *data_out* − 1) >= 0
*register* = *register* − 4 * *data_out* − 1;
*data_out* = 2 * *data_out* + 1;       % shift left and increment
else
*data_out* = 2 * *data_out*;       % shift left
end
end
*output* = de2bi(*data_out*, *data_out_size* + 1, 'left-msb');       % decimal to binary transformation
*data_out* = bi2de(*output*(1: end–1), 'left-msb') + *output*(end);  % rounding the resultend

In practice, DSP algorithms operate on various data structures, both scalars and vectors. Moreover, in the second case, it is advisable to organize parallel computations in order to reduce the total processing time, for example, through pipelining.

The proposed behavioral model has the necessary properties for organizing computations in both the sequential and parallel processing modes. The essence of parallel processing is the use of several homogeneous computing units (CU) that process the assigned data simultaneously.

The architecture of a pipeline is proposed. The continuous operating character of the pipelined module and obtaining results for different elements of the input vector *data_in* at each step are provided by compensating the calculation iterations by the number of used computing units (Figure 2); here, $N$ is the number of used CUs; $K = ||data\_in||$ is the length of the vector under processing; $j$ is the ordinal number of the vector element. The number of the model iterations for one scalar input value is (*data_out_size* + 1).

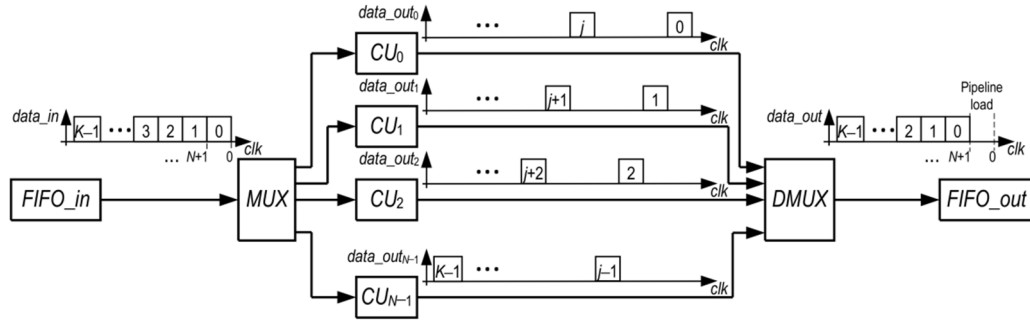

**Figure 2.** A scheme of the pipelined data processing mode.

Thus, the execution time of the OSRC for a *K*-elements vector in the sequential mode is ($K$ (*data_out_size* + 1)) clock cycles.

The use of (*data_out_size* + 1) computing units operating in parallel should compensate for the module downtime and ensure the possibility of its loading at each clock cycle at a vector processing.

Thereby, each new *i*-th input value will arrive at the *i*-th unit from (*data_out_size* + 1) available calculating units, while the previously loaded (*i*–1) values will pass further along the pipeline of the corresponding (*i*–1) computing units at each clock. The loading of the pipeline is completed at the *data_out_size* + 1 clock cycle. After that, the result of calculating the OSRC for the next element of the input vector becomes available at the output of the

pipeline at each clock. The unprocessed elements of the vector are sequentially assigned to the vacated calculating units, while the *j*-th element of the vector arrives at the *k*-th unit, where

$$k = j \, mod(data\_out\_size + 1) \tag{3}$$

The execution time of the OSRC for a *K*-elements vector in the pipeline mode is (*K* + *data_out_size* + 1) clock cycles. Thus, the use of the pipelined mode will provide an increase in efficiency in comparison with the sequential processing by a factor of

$$EF = \frac{K(data\_out\_size + 1)}{K + data\_out\_size + 1} \tag{4}$$

but it will require the hardware resources at least in (*data_out_size* + 1) times more.

## 3. Development of the IP Core on the FPGA Basis

Hardware description languages (HDL) and FPGA CAD tools provide the ability to develop functional blocks in the form of soft and hard IP cores for subsequent reuse in the designing of complex systems.

An implementation of the IP core involves the adaptation of the proposed behavioral model for the sequential and pipelined processing modes, as well as the formation of an appropriate behavioral SystemVerilog description and its synthesis for the selected FPGA basis.

The module being developed has the following configurable input parameters that ensure the versatility of the IP core, as well as the efficiency of both organizing the computational process and a set of resources occupied after synthesis, considering the initial requirements and features of the DSP problem:

*MODE* is the operating mode (sequential (slow)/pipelined (fast)), and

*TDATA_WIDTH* is the input data width (from 8 to 128 bits).

*TDATA_WIDTH* parameter allows one to set the width of the input data bus in bits and can take a value from the set {8, 16, 32, 48, 64, 128}. The IP core is parametrized by an assignment of correspondent values to the input parameters (*MODE* and *TDATA_WIDTH*) before synthesis with dependance on the solved task features and values range of operands under processing. There is a slight difference from the behavioral model: the model each time calculates the number of iterations depending on the actual value of the input number, and the module always fixes the bit width of the input data bus and, correspondingly, the number of computational iterations.

In accordance with the adapted behavioral model, the module is described in the SystemVerilog language with an orientation to the AXI4-Stream protocol [19] for signal exchange. The module has the following interface.

| The module's interface |  |
| --- | --- |
| module com_sqrt_calculator #( |  |
| parameter | *MODE* = "fast", // fast, slow |
| parameter integer | *TDATA_WIDTH* = 32 |
| )( |  |
| // Synchro signal and reset |  |
| input logic | *ACLK*, |
| input logic | *ARESETN*, |
| // Interface S_AXIS_DATA |  |
| input logic | *S_AXIS_DATA_TVALID*, |
| output logic | *S_AXIS_DATA_TREADY*, |
| input logic [*TDATA_WIDTH*–1:0] | *S_AXIS_DATA_TDATA*, |
| // Interface M_AXIS_DATA |  |
| output logic | *M_AXIS_DATA_TVALID*, |
| input logic | *M_AXIS_DATA_TREADY*, |
| output logic [TDATA_WIDTH/2:0] | *M_AXIS_DATA_TDATA* |
| ); |  |

The synthesis of both the resulting SystemVerilog description of the proposed module and the Xilinx LogiCORE ™ CORDIC IP core (hereafter Cordic IP core) [17] with similar initial settings was performed in the Xilinx Vivado® CAD system. The Xilinx FPGA SOC xc7z045ffg900-2 was used as a basis for the implementation. The results of comparing the resources being used after synthesis are presented in Tables 1 and 2, where the percentage of resources used from its total amount in the FPGA is given in parentheses. Characteristic clock cycles demonstrate the number of cycles required to calculate one square root value.

**Table 1.** The results of synthesis for the proposed module (sequential mode) and the Cordic IP core.

| FPGA Resource, Items | *TDATA_WIDTH*, Bits | | | | | | | |
|---|---|---|---|---|---|---|---|---|
| | 8 | | 16 | | 32 | | 48 | |
| | Module | Cordic IP Core | Module | Cordic IP Core | Module | Cordic IP Core | Module | Cordic IP Core |
| Logic Level | 4 | 5 | 5 | 5 | 6 | 6 | 7 | 7 |
| LUT | 25 (0.01%) | 77 (0.04%) | 45 (0.02%) | 157 (0.07%) | 69 (0.03%) | 389 (0.18%) | 97 (0.04%) | 717 (0.33%) |
| LUTRAM | 0 (0%) | 9 (0.01%) | 0 (0%) | 21 (0.03%) | 0 (0%) | 45 (0.06%) | 0 (0%) | 69 (0.1%) |
| Flip-Flop | 35 (0.01%) | 93 (0.02%) | 56 (0.01%) | 201 (0.05%) | 97 (0.02%) | 514 (0.12%) | 137 (0.03%) | 954 (0.22%) |
| Clock Cycles | 5 | 5 | 9 | 9 | 17 | 17 | 25 | 25 |

**Table 2.** The results of synthesis for the proposed module (pipelined mode) and the Cordic IP core.

| FPGA Resource, Items | *TDATA_WIDTH*, Bits | | | | | | | |
|---|---|---|---|---|---|---|---|---|
| | 8 | | 16 | | 32 | | 48 | |
| | Module | Cordic IP Core | Module | Cordic IP Core | Module | Cordic IP Core | Module | Cordic IP Core |
| Logic Level | 4 | 5 | 5 | 5 | 6 | 6 | 7 | 7 |
| LUT | 49 (0.02%) | 77 (0.04%) | 138 (0.06%) | 157 (0.07%) | 391 (0.18%) | 389 (0.18%) | 797 (0.36%) | 717 (0.33%) |
| LUTRAM | 5 (0.01%) | 9 (0.01%) | 13 (0.02%) | 21 (0.03%) | 29 (0.04%) | 45 (0.06%) | 45 (0.06%) | 69 (0.1%) |
| Flip-Flop | 76 (0.02%) | 93 (0.02%) | 174 (0.04%) | 201 (0.05%) | 490 (0.11%) | 514 (0.12%) | 966 (0.22%) | 954 (0.22%) |
| Clock Cycles | 5 | 5 | 9 | 9 | 17 | 17 | 25 | 25 |

Based on the results of the synthesis, we can conclude that the implemented module in a pipelined mode with 8- and 16-bit input data buses takes less resources than the Cordic IP core and a comparable number of resources with 32- and 48-bit input data buses. So, with an 8-bit input bus, the gain in resources is: 28 LUTs (36.4%), 4 LUTRAMs (44.4%), and 17 FFs (18.3%); with 16-bit input bus: 19 LUTs (12.1%), 8 LUTRAMs (38.1%), and 27 FFs (13.4%).

The module in the sequential mode takes up significantly less resources than the Cordic IP core, regardless of the width of the input data bus. So, with an 8-bit input bus, the gain in resources is 52 LUTs (67.5%), 9 LUTRAMs (100%), and 58 FFs (62.3%); with a 16-bit input bus, it is 112 LUTs (71.2%), 21 LUTRAMs (100%), and 145 FFs (72.1%); with a 32-bit input bus, it is 320 LUTs (82.3%), 45 LUTRAMs (100%), and 417 FFs (81.1%); with a 48-bit input bus, it is 620 LUTs (86.5%), 69 LUTRAMs (100%), and 817 FFs (85.6%).

It should be noted that the width of the input data bus of the Cordic IP core [17] is limited to 48 bits, in contrast to the width of the input data bus of the proposed and

developed module, which is limited to 128 bits, which significantly expands the range of values of the processed data.

## 4. Experimental Results

The performance assessment and technical characteristics of the proposed IP core were investigated during the verification of the behavioral model and SystemVerilog description of the module, as well as testing its hardware implementation in the FPGA.

The verification plan includes checking the adequacy of the behavioral model in the MATLAB tool using input test sets, obtaining the results of the OSRC and their subsequent analysis.

Numerical values correspond to the boundary values for the selected input bit width, as well as values with a special structure reflecting a single, double, and four-fold alternation of zeros and ones in the number code, and the exact results of the integer square root corresponding to them were used as input test sets. The results obtained during experimental study correspond to the golden reference values.

The second step of verification is aimed at checking the SystemVerilog description of the proposed module using the Synopsys VCS CAD system. The verification plan at the second step includes a functional check of the module on the same test patterns (Table 3) for both modes of operation, checking the data integrity and correctness of the module interface, as well as evaluating the results of synthesis for the selected FPGA basis.

**Table 3.** Maximum system frequency of the IP core determined after synthesis.

| *TDATA_WIDTH*, Bits | *CLK*, MHz | | |
|:---:|:---:|:---:|:---:|
| | *MODE* = "Slow" | *MODE* = "Fast" | Cordic IP Core |
| 8 | 641 | 641 | 477 |
| 48 | 513 | 508 | 339 |
| 128 | 409 | 395 | - |

Maximum clock frequency (*CLK*) of the proposed IP core and the occupied re-sources in the FPGA were determined after synthesis for various values of the tunable parameters *MODE* and *TDATA_WIDTH* (Table 3).

An experimental setup represented in Figure 3 was used for a field test of the implemented IP core and the Cordic IP core. The setup consists of a special board connected to a PC by a JTAG cable, which is used for FPGA configuration, input data application, and output data acquisition. The board is based on the Xilinx FPGA SOC xc7z045ffg900-2. The synthesized configuration file was loaded into the Xilinx FPGA SOC xc7z045ffg900-2, and functional testing of the developed IP core as part of the digital signal processing module for the 128-bit input data bus was carried out for both the sequential and pipelined modes of operation.

The input test patterns were loaded into FPGA block memory. The IP core operating results were verified automatically in the chip. The timing diagrams of the module's operation were recorded using the Xilinx ILA kernel. The proposed IP core demonstrated the correct operation at a system frequency of 330 MHz and 340 MHz for pipelined and sequential modes correspondingly. So, the difference in operating frequency between the results of synthesis and after IP-core implementation in the FPGA consists of less than 17%.

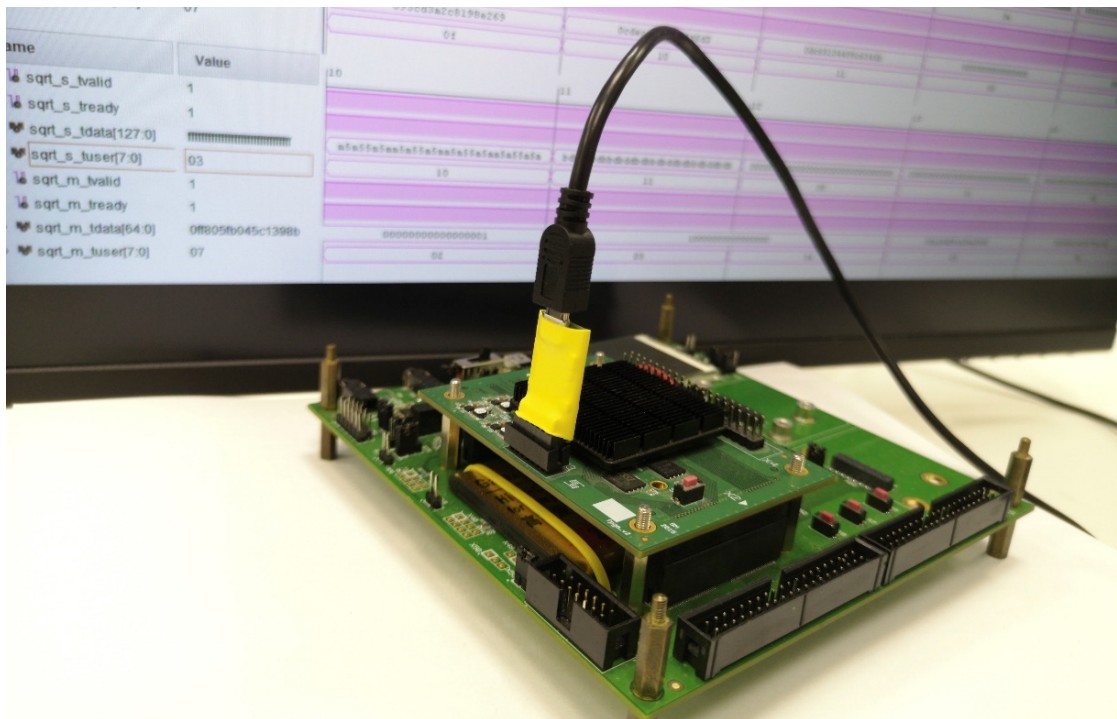

**Figure 3.** An experimental setup for the field testing.

## 5. Discussion

A non-restoring algorithm for integer square root calculation was chosen based on the minimax criterion. A behavioral model was developed in the MATLAB tool, which has the necessary properties for organizing computations in both serial and parallel processing modes. The proposed model is used to implement a configurable module on the FPGA basis, such as the IP core, operating in two modes: sequential and pipelined. The choice of the operating mode, as well as the ability to configure the input data bus width, ensure the possibility to control the number of the occupied FPGA resources.

A significant gain was revealed in resource utilization and in the maximum value of the system frequency after synthesis of the developed IP core in comparison with the corresponding integer square root IP core from the Xilinx LogiCORE ™ CORDIC library.

There are the following resource gain results: in serial mode with a 48-bit input bus, the results are 620 LUTs (86.5%), 69 LUTRAMs (100%), and 817 FFs (85.6%); in pipelined mode with a 16-bit input bus, the results are 19 LUTs (12.1%), 8 LUTRAMs (38.1%), and 27 FFs (13.4%). The gain in the maximum system frequency is 174 MHz in sequential mode with a 48-bit input bus and 169 MHz in pipelined mode. The developed IP core was verified in the Synopsys VCS CAD tool. The field test of the IP core implemented in the Xilinx FPGA SOC xc7z045ffg900-2 has demonstrated the correct operation at a system frequency closest to the theoretical one obtained after synthesis. The developed IP core for square root calculation ensures effective hardware acceleration at scalar and vector data processing and demonstrates the following benefits: high operating frequency, minimal number of occupied general resources in the FPGA SoC, and adjustment on the serial or pipelined operating mode with operands of up to 128 bits in length. Due to the low overhead in general of FPGA's resources, the implemented IP core is effective for use in designing specialized telecommunication systems based on a programmable logic, for example, software-defined radio, in which the square root operation is often used, while the specialized resources (DSP, BRAM, etc.) can be allocated to implement complex digital signal processing operations.

**Author Contributions:** Conceptualization, V.M. and S.M.; methodology, S.M.; software, V.M. and S.V.; validation, V.M., S.V. and S.M.; formal analysis, V.M.; investigation, V.M., S.V. and S.M.; resources, V.M. and S.M.; data curation, V.M.; writing—original draft preparation, V.M. and S.M.; writing—review and editing, S.M.; visualization, S.V.; supervision, S.M. All authors have read and agreed to the published version of the manuscript.

**Funding:** This paper has been supported by the Kazan Federal University Strategic Academic Leadership Program ("PRIORITY-2030").

**Data Availability Statement:** Not applicable.

**Conflicts of Interest:** The authors declare no conflict of interest.

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
