# Peer review of "A Configurable IP Core for Calculating the Integer Square Root for Serial and Parallel Implementations in FPGA"

_electronics, doi:10.3390/electronics11152335_

Round 1

Reviewer 1 Report

Apart from minor spelling checks, the paper does present a solution to one of the common issues of implementing floating point math.

It would be great if author also provides information on number of clock cycles proposed IP and Cordic IP takes for similar bus width. I found this part missing.

Author Response

Response to Reviewer 1 Comments:

Thank you very much for the valuable comment.

It would be great if author also provides information on number of clock cycles proposed IP and Cordic IP takes for similar bus width. I found this part missing.

Answer: The corresponding information about clock cycles for proposed IP and Xilinx Cordic IP has been included in Tables 1 and 2.

Reviewer 2 Report

An effective algorithm of integer square root is proposed, and a behavioral model based on a non-restoring algorithm is presented, the system allows to change the width of the input data bus. Furthermore, the results demonstrate that the gain in maximum system frequency is 174 MHz with 48-bit input bus. There are several problems needed to be addressed,

1 in the part of experiment, the comparison to other models should be added.

2 the experimental setup including the PC and the core part should be introduced, and the figures of experimental setup should be added.

3 the potential application of research results should be discussed.

Author Response

Response to Reviewer 2 Comments:

Thank you very much for the valuable comments.
1 in the part of experiment, the comparison to other models should be added.

Answer: The experimental results of the field-test are presented for two solutions. The first one is the proposed IP-core, and the second one is the Xilinx IP-Core based on the CORDIC algorithm. The corresponding algorithm has been included and described in the revised version of paper.

2 the experimental setup including the PC and the core part should be introduced, and the figures of experimental setup should be added.

Answer: The figure with experimental setup has been included in the revised version of paper and relevant description has been represented in the text.

3 the potential application of research results should be discussed.

The strengths of the proposed IP-core and perspectives for practical use have been reflected and discussed in the conclusion.

Reviewer 3 Report

The paper deals with operation of square root calculation implementation algorithm implementation. It analyzes several implementations found in the literature and treat their disadvantages. “In one bit iteration, the algorithm computes exactly one bit of the result, so it takes 8 iterations to calculate the square root of a 16-bit integer and one iteration to compute the first bit for rounding. ” Remark: Please explain. Please all code should have comment lines and explanations. Figure 1. is either scanned or not a corresponding quality. Please correct. Also explain the figure in detail. What is the difference between k and K in the figure? Equations should be numbered and explained (line 159, 163). In line 176-179 the parameters are static and not dynamic. So should be clarified that the module is parameterized before synthesis and not after implementation! The implementation data also should be presented not only the data obtained after synthesis. Also would be well come if different implementation strategies are used and analyzed. Figure 2 and 3 quality is low. Should be improved. The filed test is influenced by the embedded system created. The experiments used/not used operating system. Note that there is a conflict between: Funding and Acknowledgment. In the last 25-30 years of FPGA technology the square root computation was in the focus of research. What are the benefits of this implementations. Is this available for the research community? The paper presented an implementation of square root algorithm. Is compared to the cordic one. The tendencies in FPGA are the use of Vitis platform instead of Vivado. What would be the benefit of extending the algorithm to adaptive computing?

Author Response

Response to Reviewer 3 Comments:

Thank you very much for the valuable comments.
1 “In one bit iteration, the algorithm computes exactly one bit of the result, so it takes 8 iterations to calculate the square root of a 16-bit integer and one iteration to compute the first bit for rounding. ” Remark: Please explain.

Answer: The algorithm is based on sequential consideration of a pair of the operand bits, so at each iteration step one bit of the result is formed, and the number of iterations is finite and deterministically equal to half the length of the operand to calculate the integer part of the square root, an additional iteration is used to calculate the rounding bit. The corresponding changes have been included in the revised version of paper.

2 Please all code should have comment lines and explanations.

Answer: Thank you. Comment lines and explanations have been included.

3 Figure 1. is either scanned or not a corresponding quality. Please correct. Also explain the figure in detail. What is the difference between k and K in the figure?

Answer: Figure 1 was blurred after converting to PDF. Original picture was generated according to requirements. Corresponding explanations for variables used in Figure 1 have been included in the revised version of paper.

4 Equations should be numbered and explained (line 159, 163).

Answer: All equations have been enumerated and used variables explained.

5 In line 176-179 the parameters are static and not dynamic. So should be clarified that the module is parameterized before synthesis and not after implementation!

Answer: You are right. The correspondent comments have been included in order to clarify the need of parametrization before synthesizing of IP-core.

6 The implementation data also should be presented not only the data obtained after synthesis. Also would be well come if different implementation strategies are used and analyzed.

Answer: The characteristics of the proposed IP-core after implementation was represented in the text for the pipelined and sequential modes. 

7 Figure 2 and 3 quality is low. Should be improved.

Answer: According to suggestions of other reviewers, Figures 2 and 3 have been removed from the revised version of paper.

8 The filed test is influenced by the embedded system created. The experiments used/not used operating system.

Answer: The field test studied the independent characteristics of the proposed IP-core using the online calculation of square-root and offline initial data assignment and results acquisition. Therefore, nothing operating systems used during an operation frequency estimation.

9 Note that there is a conflict between: Funding and Acknowledgment.

Answer: The author’s team did not receive a financial support by any grant or foundation but used technical facilities and software in the framework of university support program. Therefore, we have decided to express an acknowledge to the university and corresponding program.

10 In the last 25-30 years of FPGA technology the square root computation was in the focus of research. What are the benefits of this implementations. Is this available for the research community?

Answer: The strengths of the proposed IP-core and perspectives for practical use have been reflected and discussed in the conclusion.

11 The tendencies in FPGA are the use of Vitis platform instead of Vivado. What would be the benefit of extending the algorithm to adaptive computing?

Answer: Vitis platform is used in the cases of a MPSoC design combining the development of programmable logic devices (FPGA) and software for internal CPU. Such approach and the tool are used for software-centric design. Vivado in their turn is used for hardware-centric design. In our case, the IP-Core (pure hardware solution) is the object under consideration and implementation. Therefore, Vivado CAD tool was used in the design flow. The proposed IP-core takes less FPGA resources and operates on high frequencies for scalars and vectors processing. These features make the proposed IP-core more attractive for use in design of complex systems because more of specific resources (DSP, BRAM, etc.) are available for implementation of efficient algorithms including adaptive computing algorithms.

Reviewer 4 Report

The paper presents the implementation of an algorithm for computing square roots in hardware. The choice of the algorithm was based on a complexity analysis and demonstration of the algorithm's strengths for an FPGA implementation. The algorithm was implemented in an IP-core format for Xilinx FPGAs and presents good performance and resource usage results when compared to the IP-core provided by Xilinx.

Strengths:

The work stands out in the implementation and testing of IP-core hardware developed.

Weaknesses:

The algorithm is the same shown in [15] A new non-restoring square root algorithm and its VLSI implementations. There is no novelty, unless a FPGA implementation.

The work presents a comparison with another IP-core provided by Xilinx, but details of the algorithm used in this IP are not discussed. The work presents only one implementation of an algorithm although others are cited, the implementation of these other algorithms could better demonstrate why the chosen algorithm is better. The figures with the circuit waves are not legible and do not help in understanding the work.

Questions:

  1. The Cordix seems to be already a pipeline circuit. It is not fair to compare a sequential size to a pipeline implementation as shown in Table 1.

  2. Please, remove the waveforms and table 3. It is an academic work, and this is not add any new information. 

  3. Please give more details regarding the Cordix Xilinx implementation and algorithm.

  4. Currently FPGAs have large amounts of DSPs and memory blocks, couldn't these resources be used to improve IP performance?

  5. Why have other algorithms not been implemented in order to demonstrate the advantages and disadvantages of each one?

  6. All results are simulation. Please add a case study that includes a FPGA board execution to validade a large data stream includes input/output to CPU or external FPGA memory. 

  7. Please rewrite the comparison to Cordix, it is a bit confusing “620 LUTs (86.5%),” in table 2 module has 797 and Cordix has 717. Please clarify it.

Author Response

Response to Reviewer 4 Comments:

Thank you very much for the valuable comments.

1 The Cordix seems to be already a pipeline circuit. It is not fair to compare a sequential size to a pipeline implementation as shown in Table 1.

Answer: You are right. The Xilinx Cordic library has two architectural configurations the word series and parallel. Meanwhile the Xilinx IP-Core for the square root calculation is synthesized for the parallel architecture only. Therefore, the comparative analysis of obtained results was done with only one configuration of the Xilinx Cordic IP-Core.

2 Please, remove the waveforms and table 3. It is an academic work, and this is not add any new information.

Answer: Ok. Figures 2 and 3 as well as Table 3 have been removed from the text.

3 Please give more details regarding the Cordix Xilinx implementation and algorithm.

Answer:

Corresponding part has been included in Section 2 as a subsection.

4 Currently FPGAs have large amounts of DSPs and memory blocks, couldn't these resources be used to improve IP performance?

Answer:

You are right. Meanwhile, the state-of-the-art FPGAs includes special internal blocks as DSP, BRAM, etc. but the number of such specific resources essentially less in comparison with general LUTs, FFs, LUTRAM. These specific blocks can be reasonably use for implementation complex digital signal processing operations, therefore the main goal of the proposed in paper IP-core deals with optimal use of FPGA internal resources (mostly general not specific) with reasonable accuracy and performance.

5 Why have other algorithms not been implemented in order to demonstrate the advantages and disadvantages of each one?

Answer:

The selected non-restoring algorithm was compared with CORDIC algorithm as well as technical characteristics of the proposed and implemented IP-core were compared with characteristics of commercial the Xilinx LogiCORE™ IP CORDIC core.

6 All results are simulation. Please add a case study that includes a FPGA board execution to validade a large data stream includes input/output to CPU or external FPGA memory.

Answer: Indeed, the simulation was used for behavioral model only. A SystemVerilog-description of the proposed module was verified by special the Synopsys CAD tool, and IP-core implemented in FPGA was physically tested. So, all stages of the design flow from a model development up to device functional testing are represented in the paper. A photo of used FPGA board has been included in the revised version of paper instead of waveform diagrams.

7 Please rewrite the comparison to Cordix, it is a bit confusing “620 LUTs (86.5%),” in table 2 module has 797 and Cordix has 717. Please clarify it.

Answer: The values in lines 193-204 represent the difference between number of used corresponding resources in the Xilinx IP-Core and proposed IP (the gain in resources, not absolute values). Phrase “620 LUTs (86.5%),” corresponds to the sequential mode (Table 1) and is correct.

Round 2

Reviewer 2 Report

no more comments

Reviewer 4 Report

All comments have been addressed.